# The Effects of *Lactobacillus johnsonii* on Diseases and Its Potential Applications

**DOI:** 10.3390/microorganisms11102580

**Published:** 2023-10-17

**Authors:** Ziyi Zhang, Lanlan Zhao, Jiacheng Wu, Yingmiao Pan, Guoping Zhao, Ziyun Li, Lei Zhang

**Affiliations:** 1Microbiome-X, School of Public Health, Cheeloo College of Medicine, Shandong University, Jinan 250000, China; szhangziyi@163.com (Z.Z.); zhaollxm@126.com (L.Z.); jiacheng_wu@mail.sdu.edu.cn (J.W.); pym16cg@163.com (Y.P.); gpzhao@sibs.ac.cn (G.Z.); 2State Key Laboratory of Microbial Technology, Shandong University, Qingdao 266000, China; 3CAS Key Laboratory of Computational Biology, Bio-Med Big Data Center, Shanghai Institute of Nutrition and Health, University of Chinese Academy of Sciences, Chinese Academy of Sciences, Shanghai 200000, China

**Keywords:** probiotics, *Lactobacillus johnsonii*, diseases, antimicrobials

## Abstract

*Lactobacillus johnsonii* has been used as a probiotic for decades to treat a wide range of illnesses, and has been found to have specific advantages in the treatment of a number of ailments. We reviewed the potential therapeutic effects and mechanisms of *L. johnsonii* in various diseases based on PubMed and the Web of Science databases. We obtained the information of 149 *L. johnsonii* from NCBI (as of 14 February 2023), and reviewed their comprehensive metadata, including information about the plasmids they contain. This review provides a basic characterization of different *L. johnsonii* and some of their potential therapeutic properties for various ailments. Although the mechanisms are not fully understood yet, it is hoped that they may provide some evidence for future studies. Furthermore, the antibiotic resistance of the various strains of *L. johnsonii* is not clear, and more complete and in-depth studies are needed. In summary, *L. johnsonii* presents significant research potential for the treatment or prevention of disease; however, more proof is required to justify its therapeutic application. An additional study on the antibiotic resistance genes it contains is also needed to reduce the antimicrobial resistance dissemination.

## 1. Introduction

Probiotics are correctly defined as “live microorganisms that, when administered in adequate amounts, confer a health benefit on the host” [1]. *Lactobacillus johnsonii*, as one of the typical intestinal probiotics, is widely distributed in the gastrointestinal tracts (GITs) of several hosts, including humans, mice, dogs, poultry, pigs, and honeybees [2,3,4], and has a long history of application in the food and fermented feed industries [5,6]. With the rapid development of science and technology, *L. johnsonii* has also been recognized as having important applications in many fields such as biology, agriculture, animal husbandry [7], and medicine [8]. Many studies have been conducted on *L. johnsonii* to explore its specific function and mechanism in different diseases, such as colitis, diarrhea, liver disease, and so on. In animal models and human models, researchers have conducted relevant studies, and it should be noted that a large number of studies show that *L. johnsonii* exhibits the following beneficial abilities: anti-inflammatory, immunomodulatory, intestinal microflora balance, and intestinal barrier protection. Moreover, *L. johnsonii* co-evolved with different animals at the species or strain level [9,10], which provides a reasonable basis for speculating on its relationship with health benefits. *Lactobacilli* represent the types of microorganisms to which the mammalian immune systems have learned not to respond, and this is considered a potential driver for the evolution of the human immune system [9]. According to a substantial body of literature, *L. johnsonii* has been shown to play a crucial role in modulating the host immune system, by altering macrophage [11], T-cell, and Th2 cytokine levels [12,13] and regulating dendritic cell (DC) function [14].

As of 14 February 2023, we retrieved a total of 1313 results in the Web of Science using “*Lactobacillus johnsonii*” as a keyword, including 1194 papers, 180 reviews, 46 clinical trials, etc. Although there were studies on the beneficial effects of *L. johnsonii* on certain diseases, we found that there was no review that comprehensively summarized the potential beneficial effects of *L. johnsonii* on different diseases to date, and the role *L. johnsonii* plays in disease treatment is unclear. This review summarizes the potential beneficial effects of different strains of *L. johnsonii* in a variety of common diseases involving various parts of the body and is useful for other researchers to quickly understand the field and to conduct more refined studies.

## 2. Comprehensive Characteristics of Identified *L. johnsonii*

We searched the National Center for Biotechnology Information (NCBI) and overviewed information tables for the *L. johnsonii* (as of 14 February 2023) (Appendix A). From Appendix A, it is easy to find that these strains were identified through shotgun metagenomic sequencing from host samples, while others were isolated from the host samples. The strains isolated from the hosts came from different body parts of different hosts in different countries, including the human intestine, mouse forestomach, and pig intestine. Based on the available information, we can know that the earliest strains were collected in 1964. However, the culture conditions required for many strains were not described in detail.

Plasmids, as genetic units in the bacterial cytoplasm independent of chromosomes, facilitate bacterial growth. We searched the name of the strain in NCBI, utilizing “Nucleotide” as the search database. In the record page, we were able to access the strain-related plasmid information, which contains the name, description, sequence, and other relevant details of the plasmid, and we summarized the relevant content to obtain the information in Table 1. We found that 9 of these 149 strains contained plasmids, including DC22.2 which contained four plasmids.

High rates of antibiotic resistance were found in multiple *Lactobacill* species [15,16,17,18], including *L. johnsonii* [17]. Among them, *tet(W/N/W)* are the most widely distributed ARGs in *L. johnsonii* [18]. However, through an extensive literature search, we found that not much research has been done on the antibiotic resistance of *L. johnsonii*, which means that more research is needed to characterize the antibiotic resistance of *L. johnsonii* and to investigate the mechanisms of resistance and the possibility of transmission.

**Table 1 microorganisms-11-02580-t001:** Summary of plasmid prevalence in *L. johnsonii*.

Strain	BioSample	Size (Kb)	Replicons	CDS	Release Date
FI9785	SAMEA2272487	3.55	p9785S:NC_012552.1/AY862141.1 [19]	2	April 2009
FI9785	SAMEA2272487	26.27	p9785L:NC_013505.1/FN357112.1 [20]	26	November 2009
BS15	SAMN04631277	45.84	LJBSp1:NZ_CP016630.1/CP016630.1	43	August 2016
UMNLJ22	SAMN04573146	27.88	pUMNLJ22_1:NZ_CP021705.1/CP021705.1	34	June 2017
UMNLJ22	SAMN04573146	24.93	pUMNLJ22_2:NZ_CP021706.1/CP021706.1	24	June 2017
UMNLJ21	SAMN04573145	21.52	pUMNLJ21_1:NZ_CP021701.1/CP021701.1	22	June 2017
UMNLJ21	SAMN04573145	15.25	pUMNLJ21_2:NZ_CP021702.1/CP021702.1	20	June 2017
pf01	SAMN02469597	26.46	pLJPF01L:CP024782.1 [21]	0	November 2017
pf01	SAMN02469597	14.24	pLJPF01S:CP024783.1 [21]	0	November 2017
LL8	SAMN13266521	77.56	unnamed:NZ_CM019125.1/CM019125.1	73	December 2019
DC22.2	SAMN11371966	7.65	pLjDC22.2_1:NZ_CP039262.1/CP039262.1	10	January 2020
DC22.2	SAMN11371966	5.75	pLjDC22.2_2:NZ_CP039263.1/CP039263.1	3	January 2020
DC22.2	SAMN11371966	7.08	pLjDC22.1_3:NZ_CP039264.1/CP039264.1	4	January 2020
DC22.2	SAMN11371966	13.77	pLjDC22.2_4:NZ_CP039265.1/CP039265.1	6	January 2020
G2A	SAMN11618738	130.11	unnamed1:NZ_CP040855.1/CP040855.1	154	March 2020
G2A	SAMN11618738	108.72	unnamed2:NZ_CP040856.1/CP040856.1	103	March 2020
GHZ10a	SAMN16131614	13.65	unnamed1:NZ_CP062069.1/CP062069.1	15	October 2020
GHZ10a	SAMN16131614	15.79	unnamed2:NZ_CP062070.1/CP062070.1	18	October 2020

## 3. Effects of *L. johnsonii* on Different Diseases

Probiotics may affect the host through a variety of mechanisms, including enhancing the barrier effect of the intestinal epithelium [22,23,24]; regulating immune function [25,26]; producing organic acids [27], such as the production of oleic acid to play an anti-inflammatory role [28]; interacting with intestinal flora [29]; and interacting with the host through the cell surface structure [30]. Not all mechanisms have been confirmed in humans, nor do they exist in every probiotic strain [31]. The results of previous research we have collected indicate that the common mechanism of action of *L. johnsonii* in different diseases may include regulating immune function, interacting with intestinal flora, and improving barrier function (Figure 1). Table 2 summarizes some relevant studies and results in detail.

## 4. The Common Mechanism of *L. johnsonii* in Different Diseases

### 4.1. Respiratory Insults

Respiratory syncytial virus (RSV) infects nearly all infants by 2 years of age and is the leading cause of bronchiolitis in children worldwide [66]. Kei E. Fujimura et al. provided evidence that *L. johnsonii* supplementation significantly reduced the RSV-induced pulmonary responses [38], via immunomodulatory metabolites and altered immune function [14]. Their further study demonstrated that *Lactobacillus* modulation of the maternal microbiome enhanced airway protection against RSV in neonates. Their evidence was prenatal supplementation with *L. johnsonii*, which decreased inflammatory metabolites in maternal plasma and breastmilk, and offspring plasma, and resulted in a consistent gut microbiome in mothers and their offspring [12]. The experimental results of Chung-Ming Chen et al. showed that intranasal *L. johnsonii* administration improved lung development in hyperoxia-exposed neonatal mice [67].

### 4.2. Gastrointestinal Disease

*L. johnsonii* NCC 533 (first designed La1) (CNCM I-1225) (Nestlé, Switzerland), isolated from human intestinal microbiome, has been well characterized with regard to its potential antimicrobial effects against the major gastric and enteric bacterial pathogens and rotavirus [68]. *Helicobacter pylori* infections, colitis, *Escherichia coli*-induced diarrhea, and subclinical necrotizing colitis in farms were all possible results of *L. johnsonii* (Figure 2).

*L. johnsonii* La1 has been shown to exert an anti-inflammatory effect in many double-blind, placebo-controlled clinical trials as a drinkable, whey-based La1 culture supernatant [41], as acidified milk containing live La1 cells (LC-1) [39], or as a probiotic-containing dietary product [40,42] to *H. pylori*-positive asymptomatic volunteers. Dionyssios N. Sgouras et al. observed that a pronounced anti-inflammatory effect was exerted by La1 in particular on *H. pylori*-associated neutrophilic and lymphocytic infiltration [43], and a similar effect was found in *L. johnsonii* MH-68 [45]. In addition to the anti-inflammatory effect mentioned, there are other mechanisms that play a role. Some in vitro results suggest that GroEL proteins from La1 and other lactic acid bacteria might play a role in gastrointestinal homeostasis due to their ability to bind to components of the gastrointestinal mucosa and to aggregate *H. pylori* [69]. *L. johnsonii* La1 can also produce bacteriocins, which have inhibitory activity against the human gastric pathogen *H. pylori* [70], and its antibacterial activity was due to the production of lactic acid and (an) unknown inhibitory substance(s) [71]. However, it would seem highly unlikely that an actively secreted bacteriocin produced by La1 would retain activity, given the abundance of proteolytic activity present in the gastric epithelium [46]. *L. johnsonii* No. 1088, a novel strain that was isolated from the gastric juice of a healthy Japanese male volunteer, can inhibit the growth of *H. pylori* and suppress gastric acid secretion [44]. The role of such probiotic strains in the complex regulation of proinflammatory signal strength during early infection and other aspects need to be further identified.

Colitis refers to inflammatory lesions of the colon that occur for various reasons, as a broad concept, which can be subdivided into many categories, and it is a common intestinal disease. The main clinical manifestations are diarrhea, abdominal pain, mucus, and pus and blood stool, etc. Ding-Jia-Cheng Jia et al. uncovered that the abundance of *L. johnsonii* was lessened in colitis and identified that *L. johnsonii* relieved experimental colitis [49], drawing the same conclusions as Yunchang Zhang et al. [51]. Rogatien Charlet et al. also provided evidence that the mixed gavage of *L. johnsonii* and *B. thetaiotaomicron* alleviated acute colitis induced in mice [52]. In addition, *L. johnsonii* plays a role in the treatment of different *E. coli*-induced diarrhea, including enteroinvasive *E. coli* [48] and enterohemorrhagic *E. coli* [47], by modulating gut microbiota. In addition to *E. coli*-induced diarrhea, Keyuan Chen et al. demonstrated that *L. johnsonii* L531 helps to prevent *Salmonella typhimurium*-induced diarrhea in mice [72].

In the poultry industry, necrotic enteritis (NE), an enteric bacterial disease, significantly impacts the attempts to increase global poultry production, whereas the more prevalent subclinical form of NE (SNE) is usually difficult to detect, thereby causing considerable economic and profitability losses [73]. Hesong Wang et al. demonstrated in a previous study that feed supplementation with *L. johnsonii* BS15 may prevent the SNE-caused decrease in the growth performance of broilers [53]. The potential mechanisms include enhancing intestinal immunity and blood parameters related to immunity [54], decreasing fat deposition via adjusting the ratio of Firmicutes/Bacteroidetes in the gut [55], and influencing both lipid synthesis and catabolism in the liver [56]. RNA sequencing of gene expression extracted from liver samples also supported this mechanism [57]. In addition to adding *L. johnsonii* through feed, vaginal injection of *L. johnsonii* can modulate the mucosal barrier function and fallopian tube microbiota of laying hens, which may improve egg biosecurity [74].

### 4.3. Mental Health

The causes of mental health problems are complex. In recent years, many researchers have offered new insights into mental health problems from the perspective of the gut microbiome [75]. The association between the gut environment, host behavior, and potential psychobiotics/probiotics has been extensively investigated presently [76]. Studies have shown that *L. johnsonii* is a potentially beneficial bacterium that can improve memory impairment and modulate metabolism-related disorders through the brain–gut axis (Figure 3). The hippocampus is considered a crucial brain region in memory ability [77]; therefore, much of the research has focused on inducing hippocampus-related memory dysfunction in animal models and using this as a premise to identify potential psychobiotics or probiotics. In a mouse model of colitis, treatment with *L. johnsonii* restored the disturbed gut microbiome composition, lowered the gut microbiome, and attenuated memory impairment and colitis [78]. Ning Sun et al. demonstrated that *L. johnsonii* BS15 can prevent memory dysfunction induced by chronic high-fluorine intake through modulating the intestinal environment and improving gut development [59], and Jinge Xin et al. came to a similar conclusion [58]. Hesong Wang et al. concluded that *L. johnsonii* BS15 pretreatment enhanced intestinal health and prevented hippocampus-related memory dysfunction [30,31]. All of these indicate the psychoactive effects of *L. johnsonii* BS15 on positively influencing the brain–gut axis. In the description of mechanisms on how *L. johnsonii* BS15 yields positive psychiatric effects in psychopathology through the brain–gut axis, they all mentioned the intestinal barrier protective effects of this potential psychobiotic. The present results show that *L. johnsonii* BS15 pretreatment can reduce levels of TNF-α, IFN-γ, and IL-1β in the small intestines of mice. This result indicates the ability of *L. johnsonii* BS15 to protect the intestines from inflammation (development, digestive enzyme activities, and anti-inflammatory level). The results show that *L. johnsonii* BS15 can inhibit proinflammatory cytokines (TNF-α, IFN-γ, and IL-1β) or increase anti-inflammatory cytokines (IL-4 and IL-10) to maintain intestinal integrity [26,27].

### 4.4. Obesity

In the above reference to SNE, it was noted that *L. johnsonii* can decrease fat deposition in broiler chickens. This suggests to us that it may also have some beneficial effects on obesity. For rats on a high-fat diet, non-viable *L. johnsonii* JNU3402 (NV-LJ3402) [63], *L. johnsonii* N6.2, and blueberry phytophenols [62] can help correct diet-induced dyslipidemia. Another strain, *L. johnsonii* BFE6154, was also proved to protect against diet-induced hypercholesterolemia through the regulation of cholesterol metabolism in the intestine and liver [64]. In another species, Shaziling pigs, Jie Ma et al. found similar results, namely that *L. johnsonii* could promote lipid deposition and metabolism [79]. As obesity is a possible risk factor for diabetes, there are also some studies that have targeted diabetes, and found that a multi-strain probiotic supplement including *L. johnsonii* MH-104 [37], *L. johnsonii* MH-68 [36], and *L. johnsonii* N6.2 [33] can reduce diabetes in rats by reducing inflammation and other aspects [34,35,80,81].

### 4.5. Liver Diseases

There has been a rise in the prevalence of nonalcoholic fatty liver disease (NAFLD) and its more advanced stage, nonalcoholic steatohepatitis (NASH), and this rising disease prevalence will cause an increase in the number of patients with cirrhosis and end-stage liver disease [82]. Insulin resistance, mitochondrial dysfunction, and oxidative stress may all play a role in the disease’s pathogenesis [36,37]. Furthermore, NAFLD can be characterized by inflammation, hepatic steatosis, and hepatocyte apoptosis [83]. Jinge Xin et al. suggested that the treatment with *L. johnsonii* BS15 may prevent diet-induced NAFLD through adjusting gut flora; improving mitochondrial dysfunction; and reducing gut permeability, serum levels of LPS and IR, and inflammation [65]. Another research study of host glycolipid metabolism noted that *L. johnsonii* NCC 533 can increase the level of GSH in the serum of mice, boost mitochondrial morphology and function in the liver, reduce hepatic lipids, and improve systemic glucose metabolism [84].

The study on the protecting mechanism of *Inonotus hispidus* against acute alcoholic liver injury additionally mentioned its ability to upregulate *L. johnsonii* abundance to safeguard mice from acute alcoholic liver injury [85]. A similar potential mechanism of action was additionally seen in the BaWeiBaiDuSan (BWBDS) protection against sepsis-induced liver injury (SILI) in mice [11].

In addition to the above diseases, there are several studies targeting the treatment of other diseases. *L. johnsonii* NCC533 (La1) has recently been shown to protect against atopic dermatitis in mice if introduced during the weaning period [32]. *L. johnsonii* 6084 alleviated sepsis-induced organ injury by modulating gut microbiota. Vazquez-Munoz et al. discovered *L. johnsonii* had excellent probiotic properties and can prevent or treat mucocutaneous candidiasis [86], especially vulvovaginal candidiasis [87], and that *L. johnsonii* UBLJ01 is a potential candidate for vaginal probiotics [88]. *L. johnsonii* has also be found to delay osteoarthritis progression [89] and alleviate the development of acute myocardial infarction [90].

## 5. Conclusions and Perspectives

Our summary of recent studies on the effects of *L. johnsonii* on different diseases shows that *L. johnsonii* acts via a variety of means, including the modulation of immune function, interaction with resident microbiota, interfacing with the host, and improving gut barrier integrity. Although multiple mechanisms are probably co-expressed in a single probiotic, the generation of any given mechanism will depend on many factors, including the physiological state of the host, etc. Despite the complexity of the gastrointestinal tract microbiome, the presence or absence of specific bacterial species can dramatically alter the adaptive immune environment and intestinal environment, such as the different strains of *L. johnsonii* mentioned above. As various links between the intestinal microbiome and other organs, termed such as the “gut-lung axis” [91], have been proposed in recent years, more attention has begun to be focused on the deeper mechanisms of disease, and a number of researchers have suggested the impact of environmental exposures on the gastrointestinal microbiome, which in turn has an impact on host immunity and thus on the development of host diseases. This has led to greater attention to microbes as an intermediate factor when considering disease.

Bacterial antimicrobial resistance (AMR) to antibiotics has dramatically increased over the past few years due to the overuse of antibiotics, among other factors, and has already reached a level that poses a significant risk to future patients [92,93]. To combat the growing problem of AMR, there is a major dearth of research and development of new antibiotics; therefore, people must turn to other alternative medicines, such as probiotic-related products and their metabolites. Furthermore, if there are numerous pertinent research studies on the potential benefits of probiotics on a particular disease, the majority of them used rat models, which are insufficient to capture the complex variety of pathogenic changes that occur throughout the progression of human diseases. To fully understand the specific potential mechanism between probiotics, intestinal microecology, and disease, in-depth research is still needed, and more large-scale clinical trials are needed to evaluate the efficacy of probiotics in the treatment of disease and the safety of probiotics in the human body. From the above situation, we emphasize that *L. johnsonii* has broad application prospects in different diseases; however, we also need to consider some of these issues. First of all, since the host specificity of *L. johnsonii* is only known at the strain level, a question to ponder is whether animal and in vitro experiments can be extrapolated to humans themselves, and deeper mechanisms need to be studied with more human samples. In addition, we found that some researchers in experimental studies used live bacteria, but some researchers used dead bacteria. This may be due to the different culture conditions suitable for different strains, and the inability of some to survive in vitro for long periods of time. Therefore, it is equally important to study the active substances of dead functional probiotics for disease prevention. Indeed, several studies have been conducted on the safety of various strains of *L. johnsonii* [94,95,96], but some studies have not been conducted on toxicity in animal models, and more strains and further studies should be conducted. In addition to the application of *L. johnsonii* alone, a study last year added weight to the possible role of probiotics and functional materials in the treatment of disease [97]. Finally, it is worth noting that our summary results showed that not much research has been done on the antibiotic resistance of *L. johnsonii*, and the mechanisms of its production and transmission have been poorly studied; therefore, more detailed and in-depth studies may be needed.

In conclusion, *L. johnsonii* still has a bright future in research, especially in the areas of using both living and dead bacteria, as well as combining different biological materials, to treat or prevent disease. Nevertheless, more clinical research is required to pinpoint the precise process in various hosts and convert the findings into practical applications.

## Figures and Tables

**Figure 1 microorganisms-11-02580-f001:**
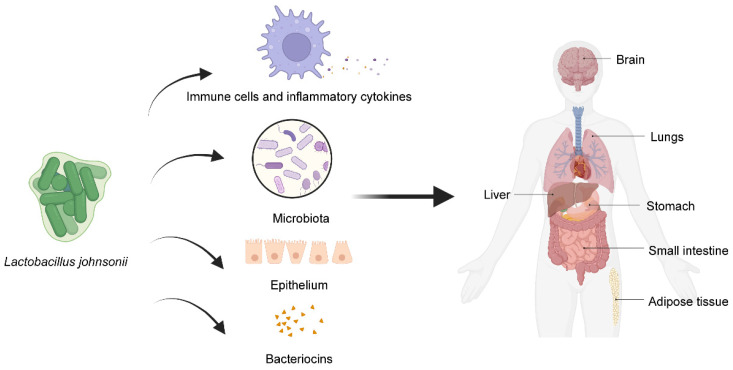
The common mechanism of *L. johnsonii* in different diseases. *L. johnsonii* acts on various parts of the body, including the brain, lungs, liver, stomach, and small intestine, as well as adipose tissue, by modulating immune function, interacting with the intestinal flora, and improving barrier functions. Created with https://www.biorender.com (accessed on 8 March 2023).

**Figure 2 microorganisms-11-02580-f002:**
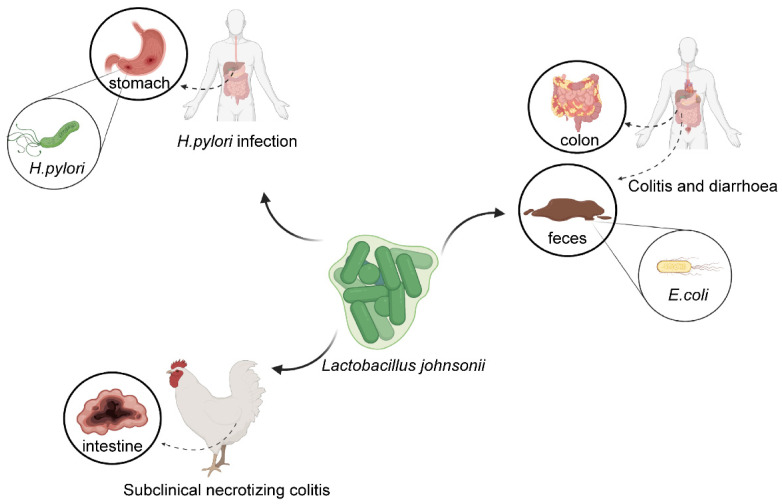
The role played by *L. johnsonii* for different gastrointestinal diseases. *L. johnsonii* is beneficial for *H. pylori* infection, colitis, *Escherichia coli*-induced diarrhea, and subclinical necrotizing colitis on farms. Created with https://www.biorender.com (accessed on 8 March 2023).

**Figure 3 microorganisms-11-02580-f003:**
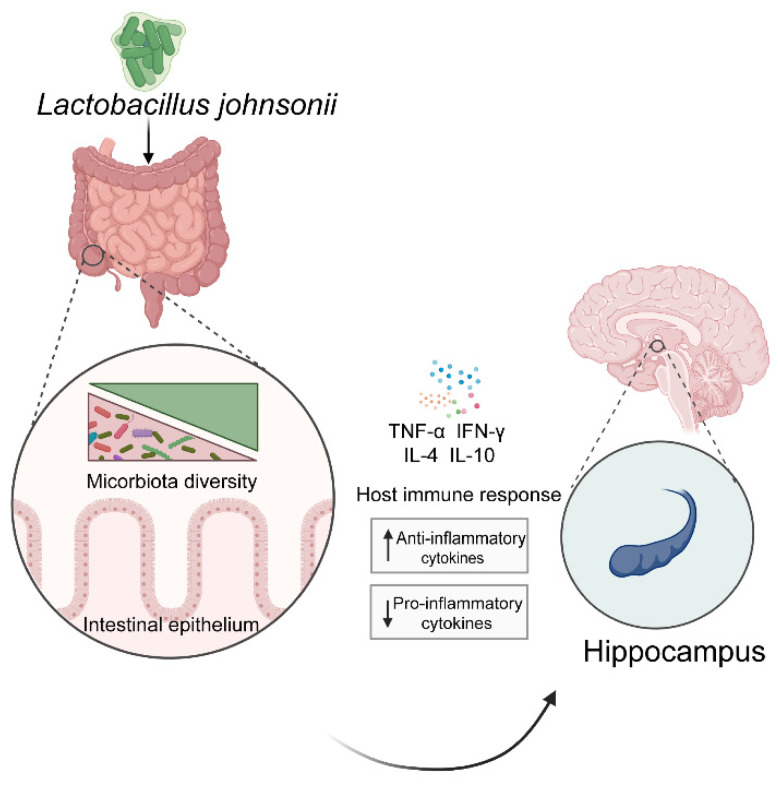
The role played by *L. johnsonii* through the brain–gut axis. *L. johnsonii* indirectly prevents hippocampus-related memory dysfunction by affecting normal gut microbes and inhibiting gut inflammatory responses. Created with https://www.biorender.com (accessed on 8 March 2023).

**Table 2 microorganisms-11-02580-t002:** *L. johnsonii* showed improvement in disease in different animal experiments or clinical studies.

Strain	Disease	Experimental Models	Duration of Intervention	Treatment Results	Reference
La1 (NCC533)	AD	Atopic dermatitis NC/Nga mice	From 20 to 22 days of age	IL-8↓, IL-12↓, IL-23↓, CD86↓	[32]
N6.2	T1D	BB-DP rats	Pre-weaning to 1 day old during mother feeding and post-weaning at 21 days old	iNOS↓, IFNγ↓, Cox-2↑, claudin↑, occludin↓	[33]
N6.2	T1D	BB-DP rats	Daily until sacrifice at diabetes onset, or the culmination of the experiment at 140 days	IL-17↑, IL-23↑	[34]
N6.2	—	BB-DP rats	After weaning to 60 days of age	mature caspase-1↓, IL-1β↓	[35]
La1	Hyperglycemia	STZ-induced diabetes animal model and hyperglycemia model induced by intracranial injection of 2DG	2 weeks	plasma glucose↓, glucagon levels↓	[34]
A multi-strain probiotic supplement including *L. johnsonii* MH-68	T1D	Patients with T1D	24 weeks	fasting blood glucose↓, HbA1c↓, IL-8↓, IL-17↓, MIP-1β↓, RANTES↓, TNF-α↓, TGF-β1↑	[36]
A multi-strain probiotic supplement including *L. johnsonii* MH-68	T2D	STZ -induced diabetes animal model	8 weeks	TNF-α↓, IL-6↓, IL-1β↓, β-cell mass↑	[37]
*L. johnsonii*	Allergic or infectious airway disease	A similar experimental design as the CRA airway challenge model and RSV infection model	—	IL-4↓, IL-5↓, IL-13↓, IL-17↓, CD11c/CD11b and CD11c/CD8, as well as CD69 activated CD4 and CD8 T cells↓	[38]
*L. johnsonii*	RSV	RSV infection model	1 week	IL-4↓, IL-5↓, IL-13↓, IL-6↓, IL-1b↓, TNFα↓, IFNβ↑, DHA↑, AcedoPC↑	[14]
*L. johnsonii*	RSV	RSV infection model	—	IL-4↓, IL-5↓, IL-13↓, IL-17↓, Gob5(mucogenic gene mRNA level) ↓Th2↓, IFN-γ↑	[12]
La1	*H. pylori*-associated gastritis	Healthy adult volunteers of both genders infected by *H. pylori*	3 weeks	—	[39]
La1	*H. pylori*-associated gastritis	Healthy adult volunteers of both genders infected by *H. pylori*	2 weeks	δ13CO2 over baseline (DOB)↓	[40]
La1	*H. pylori* infection	Healthy adult volunteers of both genders infected by *H. pylori*	2 weeks	—	[41]
La1	*H. pylori* infection	Asymptomatic school children	4 weeks	—	[42]
La1	*H. pylori*-associated gastritis	*H. pylori* SS1 strain infection mode in C57BL/6 mice	3 months	anti-*H. pylori* IgG antibody titers↓	[43]
No. 1088	*H. pylori* infection	Male germ-free Balb/c mice	2 weeks or 4 weeks	—	[44]
MH-68	*H. pylori* infection	SPF SD male rats	4 weeks	—	[45]
*L. johnsonii*	*H. pylori*-associated gastritis	Healthy adult volunteers of both genders infected by *H. pylori*	16 weeks	*H. pylori* density↓	[46]
NJ13	Enterohaemorrhagic *E. coli*-induced diarrhoea	Female mice	—	—	[47]
LJ1	Enteroinvasive *E. coli*-induced diarrhea	KM mice	8-22days	—	[48]
*L. johnsonii*	UC	DSS-induced chronic colitis mice model and human sample	—	IL10↑, TLR1/2↑, MRC1↑	[49]
La1 (NCC533)	—	Completely enterally fed elderly in-patients aged over 70 years	12 weeks	serum albumin↑, Blood Hb↑, blood phagocytic activity↑, TNF-α↓	[50]
*L. johnsonii*	Rodentium-Induced colitis	female C57BL/6J mice	2 weeks	CD4, CD8, CD11b, F4/80, TNF-α, IL-1β, IL-6, IL-17A, ssMCP1, Cox2↓	[51]
*L. johnsonii* + *B. thetaiotaomicron*	Colitis	DSS-induced chronic colitis mice model	5 days	IgA↑, IL-1β↓, IL-10↑, TLR9↑, TLR↓, MBL-C↓	[52]
BS15	SNE	Broiler chickens (Cobb 500)	days 1-28 or days 29-42	CD4↑, CD4/CD8↑, sIgA in ileum↑	[53]
BS15	SNE	Broiler chickens (Cobb 500)	28days or 42days	SOD↑, CAT↑, IHR↑, T-AOC↑, IgG↑ and IgA↑ in serum, IFN-γ↑, CD3CD4 percentage↑, CD3CD4/CD3CD8↑	[54]
BS15	SNE	Broiler chickens (Cobb 500)	4 weeks	ALT↓, AST↓, TC↓, HDL-C↑, PPARγ and ATGL↑ in adipose tissue, IGF-1 and EGF↑ in jejunum and ileum, ACC, FAS and SREBP-1c↓ inhepatic expressions, PPARα and CPT-1↑ in hepatic expressions	[55]
BS15	SNE	Broiler chickens (Cobb 500)	6 weeks	HDL-C↑, TG↓, LDL-C↓, SREBP-1c and FAS↓ in hepatic expressions	[56]
BS15	SNE induced hepatic inflammation	Broiler chickens (Cobb 500)	4 weeks	FOS↓	[57]
BS15	Fluoride-induced memory impairment	Male ICR mice	98 days	BDNF↑, CREB↑, Bcl-xl↑, Bad↓	[58]
BS15	Memory dysfunction Induced by chronic high-fluorine intake	Male ICR mice	98 days	mRNA levels of Dbn, MAP-2, and SYP↑, T-AOC, and GSH-Px↑ in hippocampuls, sIgA ↓ in the jejunal mucosa, MDA↑, SOD↑, CAT activities↑, GSH↑	[59]
BS15	Memory dysfunction in mice after RS	5C7BL/6J male mice	4 weeks	the mRNA expression levels of BDNF, CREB, SCF, c-Fos, and NMDAR↑, DA, 5-HT, and Ach levels↑, the mRNA expression level of IL-4↑, GABA↑, mRNA expression levels of bcl-2 and Bcl-xL↑	[60]
BS15	Psychological stress-induced memory dysfunction	WAS in ICR male mice	4 weeks	mRNA-expression levels of tight junction proteins claudin-1, occludin, and ZO-1 in the jejunum and ileum↑, TNF-α↓, IFN-γ↓, and IL-1β↓, mRNA levels of BDNF↑, CREB↑	[61]
N6.2 and BB	Diet-induced obesity	HFD model	15 weeks	SREBP-1↓, SCAP↓, LCFA in the serum↑	[62]
JNU3402	Diet-induced obesity	HFD model C57BL/6J mice	14 weeks	ACOX↑, CPT1↑, PGC1α↑, PPARγ↑, TG↓, FAS↓, ACC↓, SREBP1c↓, hepatic cholesterol level↓	[63]
BFE6154	Diet-induced hypercholesterolemia	HFHCD model C57BL/6J mice	4 weeks	LDL↓, ABCG8↓, NPC1L1↓, ABCG5↑, HDL↑, LDLR↑	[64]
BS15	NAFLD	HFD model ICR mice	17 weeks	LPS↓, TG↓, LDLC↓, ALT↓, FFA↓, HDLC↓, UCP-2 and cytochrome c↓ in mitochondria, the hepatic expression of Acc 1, Fas, TNFα and PPARγ↓, the hepatic expression of Fiaf↑	[65]

2DG: 2-deoxy-D-glucose; ABCG5/8: ATP-binding cassette (ABC) transporters G5 and G8; ACC: acetyl-CoA carboxylase; AcedoPC: 1-docosahexaenoylglycerophosphocholine; ACOX: acetyl-CoA oxidase; AD: atopic dermatitis; ALT: alanine aminotransferase; AST: aspartate transaminase; Bad: Bcl-xL/Bcl-2 associated death promoter; BB: blueberry phytophenols; BB-DP: BioBreeding diabetes-prone; Bcl-xl: B-cell lymphoma-extra large; BDNF: brain-derived neurotrophic factor; CAT: catalase; COX-2: Cyclooxygenase-2; CPT1: carnitine palmitoyltransferase 1; CRA: cockroach allergen; CREB: cAMP response element-binding protein; Dbn: developmentally regulated brain protein; DSS: dextran sulfate sodium; EGF: epidermal growth factor; FOS: proto-oncogene protein; FAS: fatty acid synthase; FFA: free fatty acid; GSH-Px: glutathione peroxidase; Hb: hemoglobin; HbA1c: H glycated hemoglobin; HDL-C: high-density lipoprotein cholesterol; HFD: high-fat diet; HFHCD: high-fat and high-cholesterol diet; IFN-γ: Interferon-gamma; IGF-1: insulin-like growth factors-1; IHR: inhibition of hydroxy radical; iNOS: inducible nitric oxide synthase; LCFA: long chain fatty acids; LDL-C: low-density lipoprotein cholesterol; LDLR: low-density lipoprotein receptor; LPS: lipopolysaccharide; MAP-2: microtubule-associated protein; MCP1: monocyte chemoattractant protein-1; MDA: malondialdehyde; NAFLD: non-alcoholic fatty liver disease; NPC1L1: NiemannPick C1-Like1; PGC1α: peroxisome proliferator-activated receptor gamma coactivator 1-α; PPARγ: Peroxisome proliferator-activated receptor gamma; T1D: Type 1 diabetes; T2D: Type 2 diabetes; T-AOC: total antioxidation capacity; TG: triglyceride; TGF-β1: transforming growth factor-β1; TLR1/2: toll-like receptor1/2; TNF-α: tumor necrosis factor alpha; RS: restraint stress; RSV: respiratory viral infection; SNE: subclinical (mild) necrotic enteritis; SOD: superoxide dismutase; STZ: Streptozotocin; SPF: specific pathogen free; SREBP-1: sterol regulatory element-binding protein 1; SCAP: SREBP cleavage-activating protein; SD: Sprague–Dawley; SYP: synaptophysin; UC: ulcerative colitis; WAS: water-avoidance stress; “↑” and “↓” represent the indicators of increase and decrease in the host after probiotic treatment, respectively.

## Data Availability

No new data were created or analyzed in this study. Data sharing is not applicable to this article.

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
