# Peer review of "The Effects of Lactobacillus johnsonii on Diseases and Its Potential Applications"

_microorganisms, 2023, doi:10.3390/microorganisms11102580_

Round 1

Reviewer 1 Report

Good article on the use of probiotic strains of bacteria. The abstract does not need to be abbreviated. It is better to leave it for the Introduction. Table 2 lacks information sources.

Reviewer 2 Report

In this current manuscript, the authors aimed to summarize the current literature on the effects of L. johnsonii on diseases and other related therapeutic applications. 

My main concern for the current manuscript is that it is a mixture of research updates and literature reviews. The manuscript was submitted as a review, thus including new data (Table 1, Figure 1, Table 2, and Figure S1) is not appropriate. These data need to be removed and the manuscript needs to be re-written to be considered as a review. 

Please elaborate more on the legend of Figures 2, 3, and 4. The reader should be able to understand each figure with the figure legend without going through the text. 

acceptable 

Reviewer 3 Report

The manuscript comprises two different kinds of results regarding Lactobacillus johnsonii, some of them are retrieved from the review (sections 3.4 and 3.5) , others are coming from the analysis of 149 genomes, specifically on BSH sequences and ARG (sections 3.1, 3.2, 3.3). I suggest to structure the manuscript in two different parts taking into account the two different approaches. Materials and Methods should be more detailed for the both parts. Introduction needs to highlight the importance of the species johnsonii among all the Lactobacillus species and the raisons to focuss on this. It is mention the species johnsonii was newly isolated but the references are old. The sources of isolation mentioned in S1 and in 3.1 are very diverse and disparate. Is there a logical order to present the strains in table S1 ? Could we mention host and metagenomic at the same level ? What do the authors mean for "metagenomic" , is it from the microbiote of the host ? What is the significance of ARG detection, is there the same ARG concern in human medicine ? In Fig1, what are the meaning of false / true strains carrying ARG ?, Are there multidrug resistant strains detected ? Legend is incomplet in Table 2 ( ex for ANT, LNU instead LUN ?) Are the figures 2, 3, 4 original or from publications of the review ? In this case, references must be mentioned with the figures, otherwise, these figures could be combined for a global one. Finally, the manuscript needs a lot of improvement and must be re-written.  

Author Response

请参阅附件。

Round 2

Reviewer 2 Report

The revised manuscript still contains a mixture of research results and literature reviews. For example, line 87: "we further summarized the ARGs of the existing 149 strains...". Did the authors summarize the ARGs profile based on current literature? If so, please provide references. Or did the author perform the ARG analysis in 149 genomes for this result? If this is the case, this data is original and thus cannot be included in the review. 

I strongly recommend the authors to rewrite this manuscript, either completely omitting the original data and expanding more on the literature review. OR completely rewrite the manuscript into a research paper. 

NA

Reviewer 3 Report

I thank the authors for the responses to all comments. The manuscript has been substancially improved and ready for publication
